# Long-Term Protection Against Symptomatic Omicron Infections Requires Balanced Immunity Against Spike Epitopes After COVID-19 Vaccination

**DOI:** 10.3390/vaccines13080867

**Published:** 2025-08-15

**Authors:** Heiko Pfister, Carsten Uhlig, Zsuzsanna Mayer, Eleni Polatoglou, Hannah Randeu, Silke Burglechner-Praun, Tabea Berchtold, Susanne Sernetz, Felicitas Heitzer, Andrea Strötges-Achatz, Ludwig Deml, Michaela Sander, Stefan Holdenrieder

**Affiliations:** 1Munich Biomarker Research Center, Institute of Laboratory Medicine, TUM University Hospital German Heart Center, Lazarettstr. 36, 80636 Munich, Germany; uhlig@dhm.mhn.de (C.U.); mayerz@dhm.mhn.de (Z.M.); polatoglou@dhm.mhn.de (E.P.); hannah.randeu@gmail.com (H.R.); burglechner@dhm.mhn.de (S.B.-P.); tabea-berchtold@gmx.de (T.B.); sander@dhm.mhn.de (M.S.); holdenrieder@dhm.mhn.de (S.H.); 2Occupational Medicine Service, TUM University Hospital German Heart Center, Lazarettstr. 36, 80636 Munich, Germany; sernetz@dhm.mhn.de (S.S.); heitzer@dhm.mhn.de (F.H.); stroetges@dhm.mhn.de (A.S.-A.); 3Mikrogen GmbH, Anna-Sigmund-Str. 10, 82061 Neuried, Germany; deml@mikrogen.de

**Keywords:** viral infection, predictive markers, adaptive immunity, SARS-CoV-2, vaccination, Omicron, cellular immune response, antibodies, mRNA vaccine

## Abstract

**Background**: Systematic studies providing differentiated insight into the contribution of immunity directed against conserved and non-conserved epitopes of SARS-CoV-2 Spike on long-term protection are rare and insufficiently guide future pan-variant vaccine research. The present observational cohort study aimed to evaluate the correlation of neutralizing antibody levels and cellular immunity against the Spike protein with symptomatic Omicron breakthrough infection. **Methods**: Neutralizing antibody levels against multiple (sub)variants were analyzed 6 months following the second wild-type mRNA vaccination and 6 months after booster in 107 subjects using a multiplex surrogate virus neutralization assay. To assess cellular immunity, cytokine mRNA expression levels were determined after peptide pool stimulation in whole blood samples of a study subgroup. **Results**: Neutralizing antibody titers were found to serve as a reasonably reliable correlate of protection prior to booster immunization. However, the predictive power of neutralizing antibody titers was diminished after boosting. This loss appears to be due to a critical remodeling of the antibody repertoire—a process that was dose-dependent on pre-boost humoral immunity. Vaccination against Omicron infection was most effective when a balanced immune response to both conserved and non-conserved epitopes of the viral Spike protein was induced. While neutralizing antibodies against receptor-binding domain epitopes affected by mutations were specifically associated with protection from symptomatic variant infection, cellular immunity was most effective when targeting conserved Spike epitopes. **Conclusions**: Optimal long-term protection against Omicron infection requires balanced immunity to both conserved and non-conserved epitopes of the viral Spike protein. The limited availability of cross-neutralizing antibodies targeting non-conserved epitopes and their inherently lower efficacy renders them a limiting factor as humoral immunity wanes over time. Future pan-SARS-CoV-2 variant vaccines that primarily target conserved epitopes may therefore provide less effective long-term protection against symptomatic variant infection than vaccines targeting a broader epitope spectrum including both conserved and non-conserved epitopes.

## 1. Introduction

Coronavirus disease 2019 (COVID-19) has posed a major challenge to health systems worldwide. While vaccines derived from the original strain (Wuhan-Hu-1; “wild-type”) appeared to be effective against the original strain and earlier variants, the Omicron variant displays a remarkable ability to evade neutralizing antibodies (NAbs) [1,2]. Although repeated homologous vaccinations primarily strengthen the immune response against the vaccine strain, they provide considerable protection against Omicron infection and severe clinical outcomes [3]. The increase in breadth and potency of the humoral immune response after booster vaccination depends largely on the diversification of the memory B cell pool induced already by the second vaccination [4]. Epitope masking and higher antigen availability after the second vaccination may promote B cells targeting subdominant epitopes that contribute to a broad NAb repertoire against related pathogens after booster vaccination [5].

Despite numerous mutations, particularly those acquired by Omicron, conserved epitopes on the Spike (S) protein and within the receptor-binding domain (RBD) have been identified as promising targets for vaccination strategies effective against multiple variants [6,7]. While immune escape variants are able to evade the humoral immune response to varying degrees, depending on the number and position of the mutations, CD8+ T-cell epitopes remain largely conserved among variants [8,9]. However, disease control appears to be granted by broad epitope targeting of CD4+ T cells, thereby limiting the evolution of immune evasion mutants [10].

During the observation period of the present study starting in April 2022, it was reported that more than 90% of infections in Germany were attributed to the BA.1, BA.2, BA.4, and BA.5 variants through the end of 2022 [11]. In early 2023, recombinant variants, most notably XBB.1, began to replace prevalent Omicron subvariants, reaching a relative prevalence of approximately 50% by the study’s conclusion in February. In the initial observation phase of the study, the weekly incidence was reported as 1543 cases per 100,000 citizens, dropping to a low of 75 cases per 100,000 citizens near the end of the observation phase, with intermittent spring and autumn waves peaking at 752 and 944 cases per 100,000, respectively. The median value throughout the observation period was 329 per 100,000 citizens [11].

The present study was conducted with the purpose of investigating laboratory parameters correlated with immune protection against SARS-CoV-2 infection in a single cohort of 107 healthcare workers during the 6–17 months following booster immunization with a wild-type mRNA-based vaccine. The findings of this study demonstrated the importance of a balanced immune response, covering both conserved and non-conserved epitopes, for long-term protection against variant infection. Cross-reactive antibodies targeting mutated key epitopes may become a limiting factor as antibody levels wane over time.

## 2. Materials and Methods

### 2.1. Study Design

The presented data were gathered as part of the yet unpublished Munich Observation Study of Adaptive Immune Response after COVID-19 Vaccination (MOSAIC). The MOSAIC study was initiated as an observational single-center cohort study in December 2020 at the German Heart Center, Munich, Germany. The study includes 640 voluntary employees aged between 18 and 70. Study participants received two initial doses of mRNA vaccine Comirnaty (Pfizer-BioNTech, Mainz/Berlin, Germany) followed by a single homologous booster immunization 7–13 months later. Alternatively, vector-based COVID-19 vaccine Vaxzevria (AstraZeneca, Cambridge, UK) was used instead of Comirnaty for the first one or two doses. Six months following the administration of the second and the booster dose, a blood sample was taken from each subject and analyzed for the presence of nucleocapsid (N)-antibodies, utilizing the Elecsys Anti-SARS-CoV-2 assay (Roche Diagnostics, Mannheim, Germany) and the recomWell SARS-CoV-2 IgG (Mikrogen, Neuried, Germany) in parallel. Additionally, a standardized questionnaire was used to assess the health status of the subjects at 13–17 months following booster immunization. In the event of an infection, the method of diagnosis was documented, which could be either a PCR or an antigen test from nasal and throat swabs. Subjects who were negative in all tests and reported no signs of infection may be classified as uninfected. In order to take into account the possibility of an undetected N-seronegative infection, the term asymptomatic N-seronegative is used throughout this work. Individuals who reported a history of infection between time point 2 (6 months after booster immunization; TP2) and the final assessment in February 2023 were classified as infected. Previously infected subjects were defined as those who had been infected with the Omicron variant at any point until TP2 as determined in-house by variant PCR.

In total, 107 participants of the MOSAIC study were selected for this sub-study according to the following criteria (Figure A1):(1)Vaccinated with two doses of Comirnaty between January and April 2021 followed by a single homologous booster immunization between October 2021 and January 2022. Individuals who had received the Vaxzevria vaccine were excluded from the study due to the greater diversity of vaccination schedules and the insufficient number of participants who met the other inclusion criteria.(2)No reported infection-associated symptoms (chills, cough, diarrhea, fatigue, fever, headache, joint pain, loss of smell or taste, muscle pain, sore throat, vomiting) that correlated with a positive PCR or antigen test result until last blood draw.(3)Negative test results in both SARS-CoV-2 N-antibody assays performed in our laboratory at time point 1 (TP1) and TP2.

All employees were offered the option of undergoing a PCR test in our laboratory at their own discretion in the event of a suspected infection. Additionally, all employees were provided with rapid antigen tests for home use (Clungene, Clongene Biotech, Hangzhou, China) on a voluntary basis. The results of these tests were to remain negative for the entirety of the study period (asymptomatic N-seronegative individuals) or until TP2 (infected individuals). The implementation of this procedure guaranteed the monitoring of the participants with a maximum level of intensity.

The selected subjects had a median age of 50.3 years (range: 23.1–69.9 years). The median age of infected individuals was 50.2 years (range: 23.1–69.9 years), the median age of asymptomatic N-seronegative individuals was 51.8 years (range: 34.0–66.8 years). The age of one individual was not disclosed. Overall, 67.3% of study subjects were female, constituting 64.7% of infected individuals and 71.8% of asymptomatic N-seronegative individuals.

### 2.2. Sample Collection and Infection Monitoring

Whole blood samples were obtained by forearm venipuncture and collected in lithium-heparin (for use in cell assays and Elecsys Anti-SARS-CoV-2 assay) and serum tubes (for all other antibody assays; Sarstedt, Nürmbrecht, Germany). All samples were assigned unique anonymous identification numbers and stored maximally 8 h at room temperature until use. For long-term storage, samples were frozen at −80 °C.

All commercial assays in this study were performed as single tests according to the manufacturers’ instructions. In case of invalid results, assays were repeated until the manufacturer’s specified validity criteria were met.

Two assay systems were used to detect antibodies to SARS-CoV-2: The Elecsys Anti-SARS-CoV-2 (Roche Diagnostics) is an electrochemiluminescence-based immunoassay for the qualitative in vitro detection of antibodies (not isotype specific) to SARS-CoV-2 in humans. It uses a recombinant antigen synthesized from the nucleocapsid (N) protein in a double-antigen sandwich format. The assay was performed on a Cobas E411 Analyzer (Roche). The recomWell SARS-CoV-2 IgG (Mikrogen) is an enzyme immunoassay for the qualitative and quantitative in-vitro detection of IgG antibodies against SARS-CoV-2 targeting recombinantly produced nucleocapsid protein adsorbed to a microplate. The assay was processed on a fully automated Gemini ELISA plate processor (Stratec Biomedical, Birkenfeld, Germany).

For virus detection, nasal and throat swabs were processed using an automated Maxwell nucleic acid extractor (Promega, Walldorf, Germany). The qPCR reaction was set up with an automated Mic liquid handling device (Bio Molecular Systems, Upper Coomera, Australia) using the ampliCube Coronavirus SARS-CoV-2 assay (Mikrogen). The PCR was run on a Myra qPCR cycler (Bio Molecular Systems). The same equipment was used to conduct the ampliMelt SARS-CoV-2 Variants test (Mikrogen) to differentiate virus variants.

### 2.3. Antibody Assays

Antibody levels were assessed using MesoScale Diagnostics assays (Rockville, MD, USA): RBD and Spike IgG antibodies were measured with the V-PLEX SARS-CoV-2 (IgG) assay, and neutralizing antibodies with the V-PLEX SARS-CoV-2 (ACE2) assay. All measurements were performed on a Meso QuickPlex SQ 120MM multiplex ECLIA plate luminometer (MesoScale) according to the manufacturer’s instructions. The V-PLEX SARS-CoV-2 (IgG) is a multiplex, electrochemiluminescence immunoassay (ECLIA) designed to measure IgG antibodies against multiple SARS-CoV-2 antigens (such as Spike, RBD, and nucleocapsid) in human serum or plasma. V-PLEX SARS-CoV-2 (ACE2) measures the antibodies capable of blocking the binding of labelled angiotensin-converting enzyme 2 (ACE2) to SARS-CoV-2 RBD-containing antigens. It serves as a surrogate neutralization assay to evaluate the functional activity of antibodies that block virus-receptor interaction. The assays are extensively validated and frequently used in COVID-19 research [12,13,14,15]. Wild-type and all variants of concern (VOCs) relevant during the study course were included.

The following variants and subvariants were identified through the utilization of Panel 22 and Key Variant Panel 1 product variants: (1) A (Wuhan); (2) B.1.1.7 (Alpha); (3) B.1.351 + B.1.351.1 (Beta); (4) AY.3 + AY.4 + AY.4.2 + AY.5 + AY.6 + AY.7 + AY.12 + AY.14 + B.1.617.2 + B.1.617.2-ΔY144 (Delta); (5) B.1.1.529 + BA.1 + BA.1.15 (Omicron); (6) BA.2 + BA.2.1 + BA.2.2 + BA.2.3 + BA.2.5 + BA.2.6 + BA.2.7 + BA.2.8 + BA.2.10 + BA.2.12 (BA.2 subvariants); (7) BA.2.12.1 (Omicron); (8) BA.2.75 (Omicron); (9) BA.4 + BA.5 (Omicron). For experiments representing Omicron as a whole, results of the following assays were chosen: (1) B.1.1.529 + BA.1 + BA.1.15, (2) BA.2 + BA.2.1 + BA.2.2 + BA.2.3 + BA.2.5 + BA.2.6 + BA.2.7 + BA.2.8 + BA.2.10 + BA.2.12, and (3) BA.4 + BA.5.

### 2.4. T-Cell Assays

In light of the logistical constraints and the limited availability of personnel during the ongoing COVID-19 pandemic, we proceeded to select the initial five to ten samples obtained from the daily blood draws for the planned analyses.

Within 8 h of blood collection, various SARS-CoV-2-derived peptide pools (Table A1) containing 150 pmol of each peptide were added to 500 µL of heparinized whole blood from randomly selected subjects and were incubated at 37 °C for 16 (±0.5) hours. The stimulation was terminated by the addition of RNA/DNA Stabilization Reagent for Blood/Bone Marrow (Roche) and stored at −80 °C until use. RNA was extracted with a MagNA Pure 96 Cellular RNA Large Volume Kit on a MagNA Pure 96 System (Roche). The mRNA expression of interferon-gamma (IFN-γ) and C-X-C motif chemokine ligand 10 (CXCL-10) was analyzed with T-Track SARS-CoV-2 Quant PCR (Mikrogen) on a LightCycler 480 Instrument II (Roche) using the T-Track SARS-CoV-2 Analysis Tool (version V01-00; Mikrogen) according to the manufacturer’s instructions. The T-Track SARS-CoV-2 Quant PCR allows quantification of interferon-gamma (IFN-γ) and CXCL-10 mRNA after in vitro stimulation of whole blood with SARS-CoV-2 antigens by quantitative reverse transcription PCR (RT-qPCR). It serves as indirect measure of T-cell reactivation and cellular immune response to virus-specific antigens. CXCL-10 has recently been introduced as a suitable proxy for quantifying cellular immunity against SARS-CoV-2, given its strong correlation with the activation of antigen-specific T cells [16]. Expression levels are calculated as the ratio of the stimulated and unstimulated sample, both of which have been normalized to the housekeeping marker 60S acidic ribosomal protein P0 (RPLP0). All components (except the peptide pools) and protocols were validated and approved by the manufacturer for use with or as part of an in vitro diagnostic device that received CE marking after this study.

### 2.5. Statistical Analyses

Statistical analyses were performed using Prism 9.5.1 (Dotmatics, Boston, MA, USA). Calculated *p*-values < 0.05 were considered statistically significant. To allow quantitative comparison of data from different tests, the data were normalized to a common scale using MIN/MAX normalization where indicated.

To determine if NAb values significantly decreased from TP1 to TP2, a one-tailed Wilcoxon signed-rank test was performed against a theoretical median of 0. One-tailed *p*-values were derived by halving the two-tailed *p*-values where median differences were negative.

Antibody levels and T-cell stimulation results were compared between infected and asymptomatic N-seronegative cohorts using the Mann–Whitney test (two-tailed). Variables’ correlations were analyzed using Spearman’s r.

A median line through the origin was chosen as the classification criterion for dividing the subjects into two groups in Section 3.4. The median line divides the relevant field that is shared by the subjects into two equal parts. The upper left part thus represents subjects with a higher proportion of the wild-type-specific signal than the lower right part. Statistical analyses included two-sided Fisher’s exact test for contingency tables, with Koopman asymptotic score method used to calculate relative risk confidence intervals.

## 3. Results

### 3.1. Booster Immunization Does Not Result in Sustained Enhancement of Omicron RBD Neutralization

NAb activity against multiple variants and subvariants was assessed 6 months after the second vaccination dose (TP1) and 6 months after booster immunization (TP2) by multiplex RBD-ACE2 inhibition ECLIA (Mesoscale).

The median inhibition ranged from 19.7% (Omicron B.1.1.529; BA.1; BA.1.15) to 31.3% (wild-type) in TP1 and from 14.0% (Omicron B.1.1.529; BA.1; BA.1.15) to 52.1% (wild-type) in TP2 (Figure 1a,b). Thus, booster immunization induced a marked relative median increase of individual sustained NAb levels against wild-type, Alpha (B.1.1.7), Beta (B.1.351/B.1.351.1), and Delta (variants) by 53.9% (median absolute difference with 95% CI: 19.24 [12.53, 24.47]), 50.8% (12.84 [10.33, 19.61]), 46.5% (13.56 [8.631, 17.87]), and 47.8% (12.56 [10.12, 18.12]), respectively (Figure 1c). In contrast, Omicron-specific median individual NAb levels dropped by 26.8% (B.1.1.529, BA.1, BA.1.15; −4.571 [−8.813, −2,613]), 25.5% (BA.2 subvariants; −5.882 [−8.293, −2.771]), 0.2% (BA.2.12.1; 0.03099 [−3.202, 3.184]), 35.9% (BA.2.75; −6.985 [−10.29, −3.429]), and 6.9% (BA.4, BA.5; −1.721 [−4.666, 3.314]).

TP1 levels were inversely correlated with the change in levels of all tested variants including wild-type. The correlation was strongest for the Omicron subvariants. Antibodies generated against wild-type antigen typically show lower neutralizing capacity at mutated sites compared to original sites [17]. The observed discrepancy between wild-type and Omicron variants may be explained by affinity maturation of antibodies targeting epitopes that are mutated in Omicron and the expansion of the antibody repertoire. A decline in antibodies reacting with key variant epitopes involved in ACE2-binding, even if coupled with an increase of antibodies targeting conserved sites, may lead to a net loss of variant neutralizing capacity. To assess the interval length between second vaccination and booster vaccination as a confounder on the correlation analysis of TP1 NAb levels and NAb level change, the cohort was grouped according to the interval lengths (Figure 1d–l). Almost all groups exhibited a discernible negative trend supporting an inverse correlation of TP1 levels and level change between TP1 and TP2.

### 3.2. NAb Levels Are Not Generally Suitable as Correlates of Protection

The inverse correlation between the NAb level change from TP1 to TP2 and NAb level in TP1 prompted the question of whether a reduction in NAb levels contradicts the reported initial improvement in protection against Omicron infection following booster vaccination [1]. Consequently, we correlated the data from the participants’ questionnaires on confirmed cases of COVID-19 at the end of the study with the NAb levels at TP1 and TP2.

In total, 68 of 107 (63.6%) participants were infected in the median period of 3.6 months (quartile 1: 2.0 months, quartile 3: 5.3 months, max. 9.7 months) after TP2 from June 2022 until February 2023. While not all participants were informed of their variant, we estimate that approximately 90–95% of the infection symptoms were caused by BA.1, BA.2, BA.4, and BA.5 and their subvariants and less than 10% by Delta infections, since they were by far the dominating variants throughout the observation period in Germany [11]. Given that only three individuals were infected in 2023, it can be reasonably inferred that infections with recombinant XBB.1 subvariants represent single cases at most. Overall, 51 of 68 (75%) infections were confirmed by PCR, with 15 of 51 (29.4%) being typed as Omicron and two (3.9%) being typed as Delta infections. Of PCRs, 12 (23.5%) were performed in-house and 39 (76.5%) were performed in a certified test center. In total, 17 (25%) infections were confirmed by antigen testing.

Given that antibody titers wane over time, we expected that a reduction in NAb-levels during the relevant period of 7–11 months following the last blood draw (TP2) could reduce NAb-inhibition to levels insufficient to detect clinically relevant effects in subjects with already low or medium TP2 titers. Accordingly, the study group was divided into two subgroups, characterized by low to medium (lower two-thirds of the scale) and high NAb levels (i.e., presumed effective Nab-levels; upper third of the scale) at TP2. In the subsequent experiment, the wild-type RBD NAb level was employed as the discriminator between subgroups. While overall NAb levels at TP2 did not appear to be associated with protection, TP1 levels did show some correlation, although not significantly in most cases (Figure 2). Restricting the analysis to individuals with high wild-type NAb levels at TP2 (dots in the grey box in Figure 2a and dark green circles/dark red dots in Figure 2b,d representing individuals “gated through” the grey box in Figure 2a) revealed a significant difference between infected and asymptomatic N-seronegative subjects at TP1 in wild-type (median infected (n = 22)/asymptomatic N-seronegative (n = 14) with 95% CI of difference: 32.66/44.87 [1.009, 24.27]), Alpha (29.74/40.19 [1.785, 20.73]), and Delta (23.75/36.86 [5.596, 22.30]). Obviously, the substantial number of mutations in the Omicron RBD led to the failure of the Omicron test to demonstrate significant differences between infected and asymptomatic N-seronegative individuals. The data suggest that the foundation of successful booster vaccination against symptomatic variant infection may be laid by the second vaccination. However, the effectiveness of the booster vaccination evidently depends on an additional mechanism that cannot be resolved by solely determining serum neutralizing capacity at TP2.

### 3.3. Loss of RBD-Specific IgG Antibodies After Booster Vaccination Is Associated with Symptomatic Infection

The levels of wild-type RBD-specific IgG exhibited a marked increase in TP2, with only two infected and four asymptomatic N-seronegative individuals displaying a lower level in TP2 than in TP1. When the RBD and Spike IgG levels in the datasets were grouped alternatively by low to medium or high Spike antibody levels at TP2 (in analogy to the experiment above), asymptomatic N-seronegative individuals with high Spike TP2 levels displayed higher RBD antibody levels than infected individuals, a difference that was already apparent at TP1 (median infected (n = 46)/asymptomatic N-seronegative (n = 22) with 95% CI of difference: 0.01659/0.04056 [0.005196, 0.04000]) and, to a lesser extent, at TP2 (0.6903/0.8509 [−0.004536, 0.2051]; Figure 3a,b). It was concluded that the reduction in NAb activity within the RBD, evident in Omicron variants, must be attributed to a rearrangement of the epitope repertoire of Spike-specific antibodies from TP1 to TP2.

In order to test this hypothesis, a comparative analysis was conducted between the wild-type RBD-specific and Spike-specific IgG values. A small but insignificant difference was observed between infected and asymptomatic N-seronegative individuals (Figure 3c). In contrast, infected individuals displayed a median relative loss of RBD-specific antibodies between TP1 and TP2, overt at high Spike TP2 levels, while asymptomatic N-seronegative individuals remained stable (median infected/asymptomatic N-seronegative: 0.8720/1.016; 95% CI of difference [0.004328, 0.2402]; Figure 3d). Our findings indicate that RBD-specific antibodies play a pivotal role in long-term humoral immune protection of the study cohort.

### 3.4. NAbs Against RBD Epitopes Affected by Variant Mutations Are Associated with Protection from Symptomatic Variant Infection

The marked difference between wild-type and Omicron variants in NAb activity loss suggests a shift in the ratio between conserved epitopes and epitopes affected by mutations as targets. In order to gain a deeper insight into the role of RBD-specific NAbs in protecting against variant infection, we analyzed the relationship between variant- and wild-type-specific NAbs at TP2. The correlation between the wild-type and the Alpha, Beta, and Delta variants was estimated to be approximately linear, on the basis that only a small number of mutations within the RBD can influence antibody binding [18]. The greatest degree of variation was to be anticipated for the Omicron subvariants, as they have by far the largest number of mutations within the RBD [19].

Based on a presumed dose-dependence of NAbs on the infection rate, samples with low to medium titers and high wild-type NAb titers were analyzed separately (analogous to the experiments described above). The Omicron scatterplots revealed a prominent bipolar distribution (with a left and upper “green border” formed by uninfected individuals) of infected and asymptomatic N-seronegative individuals in the group with high NAb titers (relative risk of infection of group below the diagonal line: BA.1.1.529, BA.1, BA.1.15: 2.667 [1.495, 5.535]; BA.2 subvariants and BA.4, BA.5: 2.143 [1.237, 4.201]; Figure 4). In summary, NAb levels against Omicron variants were found to be comparable in infected and asymptomatic N-seronegative subjects, as measured by the mean of three assays including BA.1, BA.2, and BA.4/BA.5 subvariants (Figure A2). Conversely, the mean difference between NAb levels directed against wild-type and NAb levels directed against Omicron variants was elevated in asymptomatic N-seronegative individuals indicating a pivotal role of mutation sites in immune protection.

**Figure 4 vaccines-13-00867-f004:**
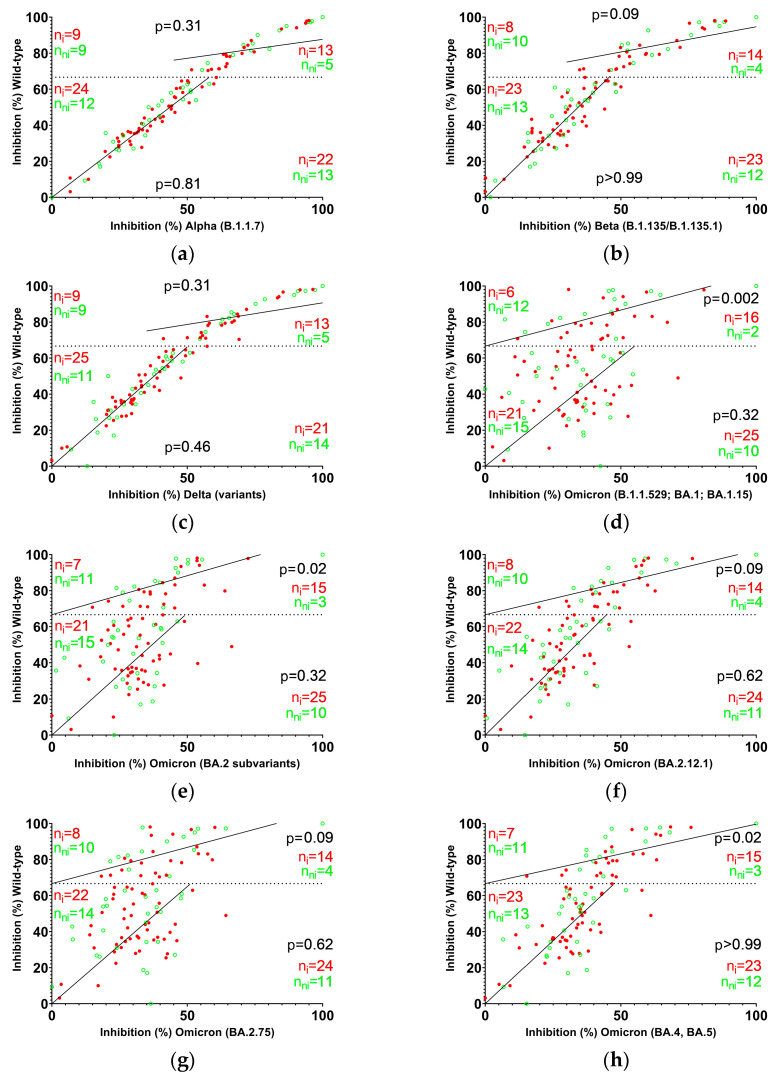
Correlation of NAb levels against wild-type and mutant RBD with infection. Min/max-normalized NAb levels ((**a**–**h**); % inhibition) at TP2 against variant and wild-type RBD, divided into high NAb and medium to low NAb groups (dotted line). The median line intersecting the graph’s origins categorizes individuals into two distinct groups. Individuals above the line are indicated as having a higher NAb level and a higher proportion of NAbs directed against wild-type antigen than individuals below the line. Counts of individuals in each group are indicated on the left (above the line) and right (below the line) in red (infected) or green (asymptomatic N-seronegative). High NAb levels with a higher proportion of NAbs directed against wild-type antigen are statistically significantly different between infected and asymptomatic N-seronegative individuals in several Omicron variants, as determined by Fisher’s exact test (results as indicated).

### 3.5. Cellular Immunity to Conserved Epitopes Is Associated with Protection from Symptomatic Variant Infection

To determine whether the humoral immune response aligns with the cellular immune response at TP2, blood cells were stimulated with peptide pools derived from wild-type and variant Spike protein and cytokine mRNA expression was assessed by RT-qPCR. The median level of CXCL-10 and IFN-γ mRNA expression in infected individuals was distinctly, but not statistically significantly, lower than that observed in asymptomatic N-seronegative individuals (Figure A3). These findings may possibly indicate that an increased number or reactivity of T cells may confer protection. A significant association between infection and a balance shifted towards non-conserved epitopes was revealed upon relating wild-type Spike peptide pools spanning exclusively mutated sites in SARS-CoV-2 variants to whole Spike (wild-type) peptide pools (median infected (n = 8)/asymptomatic N-seronegative (n = 6) with 95% CI of difference: Alpha mutation sites (CXCL-10//IFN-γ): 0.6035/0.1911 [−0.7671, −0.1917]//0.7165/0.3021 [−1.163, −0.05047]; BA.1 mutation sites (CXCL-10//IFN-γ): 1.559/0.4068 [−1.531, −0.5060]//1.142/0.3605 [−1.463, −0.2888]; Figure 5a–c). Our findings indicate that, in contrast to the humoral immune response, maintaining the balance closer to conserved-epitope-specific T cells provides superior protection from symptomatic variant infection.

It is currently unclear whether our findings regarding cellular and humoral immune protection are directly correlated. Our data depict infected and asymptomatic N-seronegative individuals with varying peptide pool stimulation ratios exhibiting ratios of wild-type to variant NAbs that are both above and below the median line. Of note, individuals infected prior to TP2 exhibited a pronounced tendency towards conserved epitopes in comparison to those infected subsequent to TP2 (median infected (n = 8)/previously infected (n = 11) with 95% CI of difference: CXCL-10: 1.559/0.9381 [−1.220, 0.01439]; IFN-γ: 1.142/0.5326 [−1.110, −0.08499]). This observation may be indicative of augmented protection against variant infection resulting from additional variant antigen exposure [20]. The use of peptide pools exclusively covering variant mutation sites and a corresponding wild-type reference pool demonstrated that there was no discernible difference in cross-reactivity with peptide variants between infected and asymptomatic N-seronegative individuals (Figure 5d–f).

In order to spatially narrow down the observed disparity between infected and asymptomatic N-seronegative individuals on the Spike protein, we conducted a separate comparison between peptide pools representing the N-terminal and C-terminal halves of all relevant Spike protein variants with the corresponding wild-type peptide pools (Figure 5g–r). To map the effect of mutations on T-cell stimulation as comprehensively as possible, the analysis was expanded to include Gamma variants. However, it can be reasonably assumed that the Gamma variant did not play a significant role in the incidence of infection during the course of this study [11]. Upon stimulation with the N-terminal peptide pool-1, the median cytokine ratio of variant to wild-type exceeded the value of 1 in five of six tested (sub)variants in infected individuals (median infected/asymptomatic N-seronegative with 95% CI of difference: Alpha pool-1 CXCL-10: 1.218/0.8041 [−1.520, −0.04511]; BA.1 pool-1 (CXCL-10//IFN-γ): 1.443/0.6259 [−1.420, −0.03581]//1.430/0.3510 [−1.503, −0.1223]; Figure 5g–r). In contrast, only two CXCL-10 ratios of the asymptomatic N-seronegative were above 1 and none of the IFN-γ ratios were. The consistent trend towards higher variant to wild-type ratios in infected individuals as compared to the asymptomatic N-seronegative localizes the source of the observed trend at least in part to the RBD-containing N-terminal half of the Spike protein. The results for peptide pool-2, which spans the C-terminal half of the Spike protein, were inconsistent. No significant difference was found for IFN-γ between infected and asymptomatic N-seronegative individuals. However, Alpha and BA.1 differed significantly in CXCL-10 values (median infected/asymptomatic N-seronegative with 95% CI of difference: Alpha pool-1: 1.323/0.6588 [−1.500, −0.02893]; BA.1 pool-1: 1.145/0.4689 [−1.277, −0.02541]), which may be due to the significant role of conserved sites on the S2 subunit and those around the S1/S2 cleavage site in protection against symptomatic infection [18,21].

A notable difference was observed in the infected groups before and after TP2, further supporting our conclusions about infection-driven T cell reactivity bias towards conserved epitopes. Cells from individuals infected with Omicron before TP2 showed cytokine ratios below 1 when stimulated with BA.1 peptides (similar to the asymptomatic N-seronegative group). In contrast, those infected after TP2 displayed ratios above 1, particularly evident with pool-1 peptides. These findings suggest that the protective effect of additional SARS-CoV-2 antigen exposure may be attributed to conserved T cell epitopes, including those in the N-terminal half of the Spike protein—specifically the RBD and N-terminal domain (NTD). This perspective is corroborated by the discovery of elevated genetic entropy levels within the T cell epitopes of the RBD and NTD, indicating viral immune evasion [21].

## 4. Discussion

The majority of the immunodominant epitopes of neutralizing antibodies against SARS-CoV-2 have undergone significant mutation in the Omicron variant, which contributes to immune evasion [22]. Immune escape can be at least partially compensated for by booster immunization [23]. This gave rise to the question of whether the booster immunization merely contributes to an increased magnitude of a broadened humoral immune response or whether it provides extended protection through the recall of specific immune adaptation processes initiated by the second vaccine dose. As demonstrated in our study, NAb levels alone are not an appropriate correlate of long-term protection after booster immunization against SARS-CoV-2 variant infection. Consequently, we sought to establish whether the augmented level of protection could be attributed to an expansion of adaptive processes that refine the repertoire of humoral and cellular immunity. Therefore, a 6-month interval was permitted after the second and third vaccination before laboratory parameters were determined.

It is important to note that individuals with a subclinical course of SARS-CoV-2 infection who failed to raise a nucleocapsid antibody response may have remained undetected. A recent study reported that 60% of confirmed SARS-CoV-2 infected cases were lacking detectable N-antibody seroconversion in mRNA vaccine recipients [24]. Similar to most previous COVID-19 studies, we cannot exclude that subclinical N-antibody negative infections may act as confounders, potentially affecting the repertoire of cellular and humoral immunity [25]. To minimize the risk of undetected asymptomatic infections, available samples were tested with two different N-antibody assays that have been approved for clinical diagnostics, with virus PCR-testing performed in case of a suspected infection, and provided participants with rapid antigen tests for home use.

On the supposition of a consistent reduction in antibody levels reported across a range of initial peak titers, our data substantiate the following interpretation of long-term immunity to Omicron variants induced by mRNA vaccination [26]:

Following the second vaccination, RBD-specific antibodies are predominantly directed towards immunodominant epitopes. The immunologic pressure (especially on immunodominant epitopes) exerted by the vaccination program and the prevalence of wild-type infections in the population drives the evolution of the virus towards the development of immune escape variants. As a consequence, Omicron RBD NAb levels are lower compared to wild-type due to the reduced number of native immunodominant epitopes. The measured differences between the Omicron subvariants and the wild-type, as well as with all other variants tested, may be attributed to the number of mutations present within the RBD (Alpha: 1, Beta: 3, Delta: 2, Omicron: 15) rather than the mutations themselves [19].

Booster immunization apparently does not provide a sufficient trigger for the expansion of immunodominant epitope-specific B cells in all individuals, resulting in inadequate compensation for the decline in functional capacity of existing plasma cells over time (Figure 1). As this outcome is dose-dependent, it could be caused by antibody-dependent immunoregulatory mechanisms such as antibody-mediated epitope masking. In numerous cases, particularly evident with the Omicron variant, this results in TP2 RBD NAb levels below the TP1 values. Elevated variant epitope NAb-levels appear to initiate an auto-regulatory response, resulting in substantial downregulation subsequent to booster vaccination. Newly formed antibodies targeting previously subdominant epitopes fail to provide equivalent quantitative compensation. The rise in median wild-type NAb-levels, concomitant with a decline in median Omicron NAb-levels, indicates that the post-booster antibody repertoire included less antibodies cross-reacting with variant epitopes. Increased efficacy at wild-type epitopes due to affinity maturation processes and an overall gain of conserved-epitope-specific NAbs may inversely account for increased wild-type NAb-levels at TP2. The reduced median NAb levels against the Omicron variant at TP2 initially seem to be at odds with the epidemiological data, which indicate an enhanced vaccine efficacy against Omicron infection following repeated immunizations [3]. However, this can be explained by elevated post-booster antibody levels targeting previously subdominant epitopes, including sites outside the RBD, which are not directly involved in ACE2 blocking [5]. These epitopes are less likely to be mutated in Omicron variants due to lower immunologic pressure prior to booster campaigns.

In the wild-type versus variant scattergrams we observed a polar distribution of infected versus asymptomatic N-seronegative subjects (Figure 4). We assume that Omicron-RBD NAb-levels mainly reflect conserved epitopes at TP2, while wild-type NAb levels represent both conserved and non-conserved epitopes. This is because numerous immune escape mutations within the RBD compromise the recognition of mutant epitopes by antibodies generated against the wild-type immunogen [22]. The far most likely explanation for differences between wild-type and mutant measurements in infected and asymptomatic N-seronegative subjects is an increased NAb level directed against non-conserved epitopes or epitopes affected by mutations. Given that RBD-specific antibodies targeting epitopes that are affected by mutations appear to be associated with protection, it seems plausible that a broad immune response, ultimately including antibodies cross-reacting with mutated sites in addition to antibodies directed against conserved epitopes, may contribute to maximum protection.

While antibody levels may be indicative of the protective efficacy of a less refined antibody repertoire, their utility is constrained following booster vaccination, as they do not account for alterations in the epitope repertoire, which appears to be a pivotal element in combating variant infections. In light of the substantial influence of both titers and the repertoire post-second vaccination on the post-booster epitope repertoire, post-booster RBD NAb levels combined with those prior to the booster offer a more accurate indicator of protection (Figure 2 and Figure 3). A shift in the balance of non-RBD- and RBD-specific Spike antibodies towards a greater proportion of non-RBD Spike antibodies may occur within 6 months after receiving a booster. This shift in antibody profile is associated with subsequent infection, which underscores the pivotal function of RBD antibodies in humoral immune protection (Figure 3).

In conclusion, a reasonable model delineating long-term immunity against symptomatic variant infection should acknowledge the fact that a number of RBD epitopes mutated in Omicron play a pivotal role in the interaction between host and virus. We thus infer that, in the context of waning antibodies over time, antibodies targeting variant RBD epitopes are the first to become a limiting factor because of the lower neutralizing capacity at mutated epitopes. This reduction in the breadth of the antibody repertoire, particularly at critical sites, may then result in increased infection susceptibility. In order to maintain a balanced humoral immunity, it may be advisable to monitor the antibody repertoire after the second vaccination and adapt antibody schedules accordingly.

Conserved epitopes appear to be a crucial element in combating symptomatic variant infection by both antibodies and T cells. Our data suggest that cellular immune protection is primarily provided by conserved epitope-specific T cells. This may be because CD8+ T cells are thought to prevent a severe course of disease rather than preventing infection by blocking virus-host cell binding [27]. However, the present study is unable to contribute to this part of the hypothesis, as it does not permit the differentiation of the role played by the two branches of the immune system in immune protection.

## 5. Conclusions

It can be finally concluded that vaccination strategies that exclusively target either conserved or non-conserved epitopes may not be an effective means of establishing an adequate level of long-term protection against infection. Consequently, new vaccines and vaccination timelines should be designed according to their specific indication. Immunity providing broad cross-variant protection appears to require conserved epitopes, while superior long-term protection may be only achieved by additionally addressing variant sites required for virus-host cell binding. Our data suggest that long-term protection is most effective when three conditions are met: (i) second vaccination achieves sufficiently high titers to trigger an efficient boost and repertoire reorganization, (ii) booster vaccination produces sufficiently high titers to counteract antibody waning, and (iii) immunity is balanced providing sufficient antibody levels targeting a broad epitope repertoire including conserved and non-conserved epitopes. Particularly in long-term immunity, antibodies cross-reacting with epitopes affected by mutations become a limiting factor as antibody levels decline over time.

## Figures and Tables

**Figure 1 vaccines-13-00867-f001:**
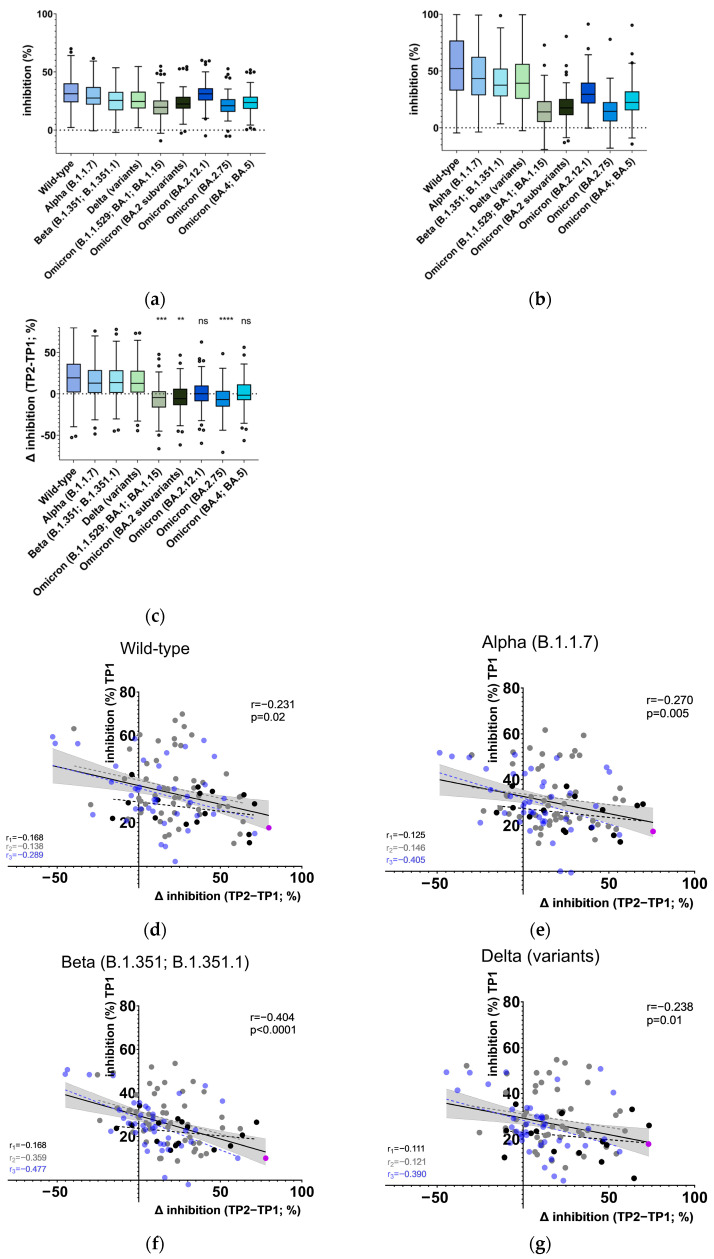
Neutralizing antibody levels after second and booster vaccination. ACE2-RBD binding inhibition in serum samples (n = 107) determined (**a**) 6 months after second vaccination or (**b**) 6 months after booster immunization; (**c**) booster immunization induced a significant overall increase in NAb levels (*p* < 0.0001) against wild-type and all variants except Omicron. Omicron NAbs decreased significantly in the majority of subvariants as determined by Wilcoxon signed-rank test (**** *p* < 0.0001; *** *p* = 0.0005; ** *p* = 0.0025; ns, not significant). Whiskers denote the 1.5 × interquartile range; (**d**–**l**) NAb levels at TP1 and NAb level changes from TP1 to TP2 are inversely correlated. Each spot represents one subject, color-coded by interval between second vaccination and booster (group 1, black: <9 months (n = 15); group 2, grey: 9 to <10 months (n = 51); group 3, blue: 10 to <11 months (n = 40); purple: >11 months (n = 1)). Inverse correlation is indicated by linear regression with 95% CI shown in grey for the entire cohort and by color coded dashed lines for each vaccination interval group. The correlation coefficient of the entire cohort and interval groups was calculated as Spearman’s r with *p*-values given in the graphs.

**Figure 2 vaccines-13-00867-f002:**
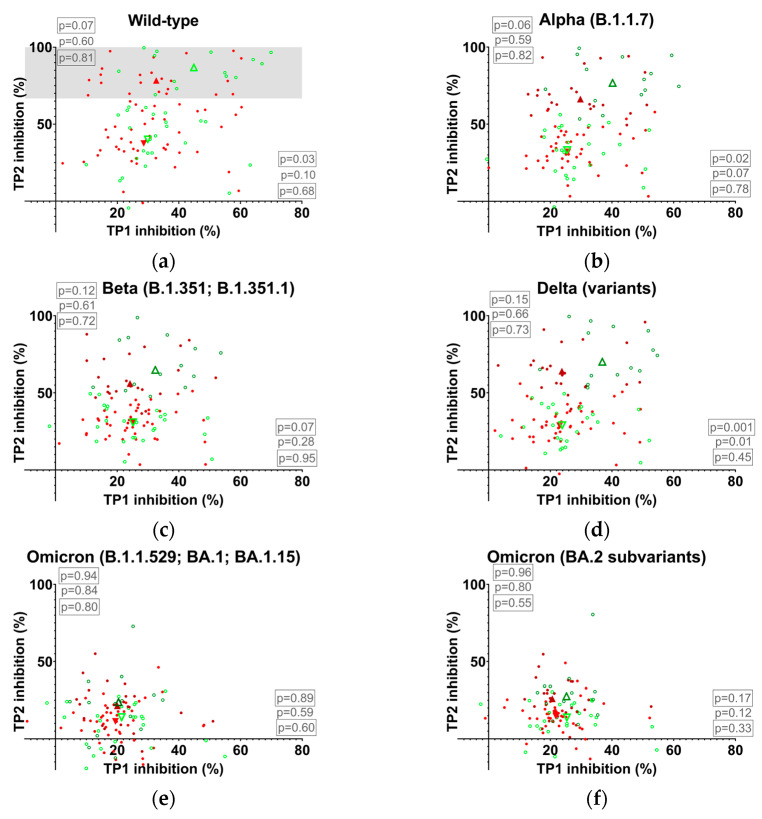
Correlation of neutralizing antibody levels after second and booster vaccination with infection. Grey box: area with high values at TP2. Triangles: median of infected/asymptomatic N-seronegative individuals from grey box in (**a**) (dark red filled/dark green with tip facing up) and median of infected/asymptomatic N-seronegative individuals from below grey box in (**a**) (red filled/green with tip facing down). Dark red dots/dark green circles: infected (n = 22)/asymptomatic N-seronegative (n = 14) individuals from grey box in (**a**). Red dots/green circles: infected (n = 46)/asymptomatic N-seronegative (n = 25) individuals from below grey box in a. *p*-values in graphs: Mann–Whitney test between infected and asymptomatic N-seronegative individuals at TP1 (right in graphs between boxed values) and TP2 (left in graphs between boxed values), and at TP1 or TP2 with datasets restricted to individuals within grey box in (**a**) (upper boxed) or values below (lower boxed).

**Figure 3 vaccines-13-00867-f003:**
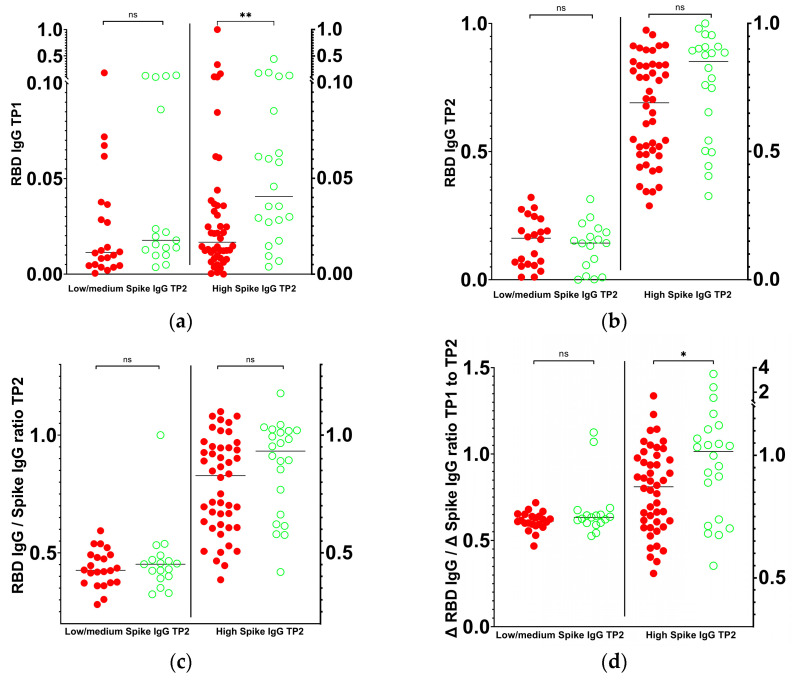
Correlation of IgG antibody levels after second and booster vaccination with infection. Min/max-normalized antibody levels directed against wild-type RBD are significantly associated with the infection rate at (**a**) TP1 and, to a lesser extent, at (**b**) TP2 when Spike-specific IgG at TP2 is high; (**c**) while RBD/Spike IgG ratios do not differ significantly between infected and asymptomatic N-seronegative individuals, (**d**) the ratio of RBD IgG to Spike IgG changes from TP1 to TP2 correlates with infection at high Spike IgG TP2 levels, as determined by Mann–Whitney test (** *p* = 0.006; * *p* = 0.04; ns, not significant). Horizontal lines: median of each group. Green circles/red dots: asymptomatic N-seronegative (high TP2 Spike IgG: n = 22; low/medium TP2 Spike IgG: n = 17)/infected (high TP2 Spike IgG: n = 46; low/medium TP2 Spike IgG: n = 22) individuals, respectively.

**Figure 5 vaccines-13-00867-f005:**
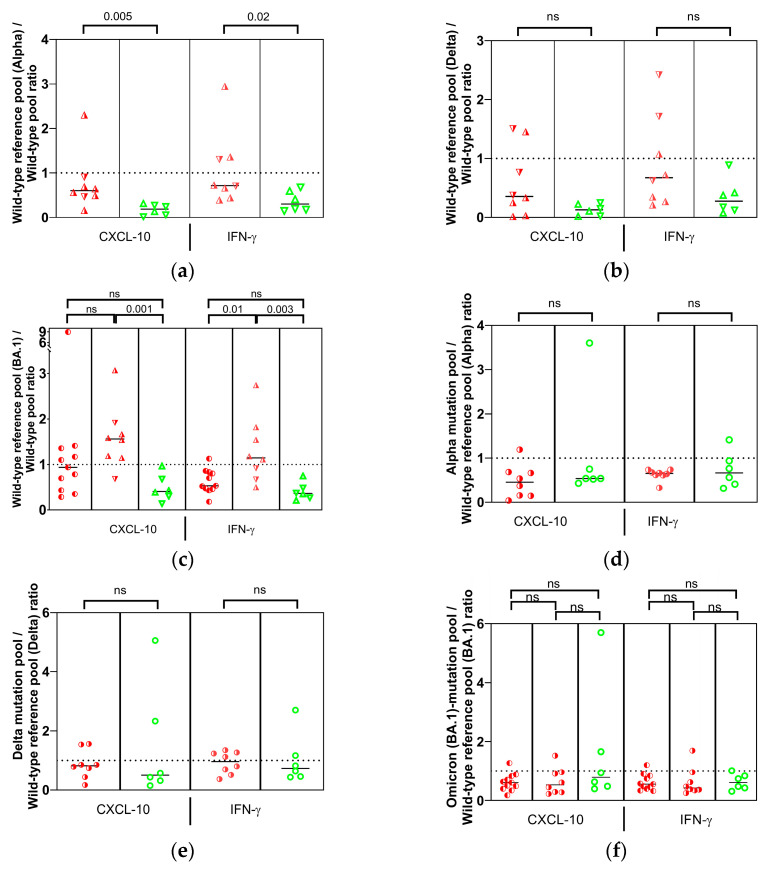
Stimulation of blood cells by wild-type and variant Spike protein-derived peptide pools at TP2 (green circles/right filled red dots: asymptomatic N-seronegative (n = 6)/infected (after TP2; n = 8) individuals); (**a**–**c**) the average ratio of cytokine mRNA expression levels between wild-type Spike peptide pools spanning exclusively mutated sites in SARS-CoV-2 variants (reference pools) to whole wild-type Spike peptide pool-1 and -2 (representing the N-terminal and C-terminal half of the Spike protein, respectively) is lower in asymptomatic N-seronegative individuals. There is no apparent association of wild-type to variant NAb ratios and T-cell stimulation (triangles tip up/tip down: individuals with NAb wild-type to variant ratio above/below median line in red, right-filled (infected after TP2) or green (asymptomatic N-seronegative); for NAb values, see Figure 4). For comparative purposes, individuals infected with Omicron before TP2 are included (left-filled red dots; n = 11); (**d**–**f**) peptide pools exclusively spanning mutated sites elicit an equivalent relative T-cell response in infected and asymptomatic N-seronegative individuals; (**g**–**r**) the stimulation ratio with peptide pool-1 and -2 is lower in asymptomatic N-seronegative, especially with pool-1. Stimulation with peptide pools revealed significant differences between infected and asymptomatic N-seronegative individuals as determined by Mann–Whitney test where indicated with *p*-values given in the graphs (ns, not significant). Horizontal lines represent the median of each group.

## Data Availability

The original contributions presented in this study are included in the Appendix A. Further inquiries can be directed to the corresponding author, subject to data protection and participant consent restrictions.

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
