# Peer review of "Long-Term Protection Against Symptomatic Omicron Infections Requires Balanced Immunity Against Spike Epitopes After COVID-19 Vaccination"

_vaccines, 2025, doi:10.3390/vaccines13080867_

Round 1

Reviewer 1 Report

Comments and Suggestions for Authors

COMMENTS

In this study, Heiko Pfiste evaluated the correlation between neutralizing antibody levels and cellular immunity against the Spike protein in relation to symptomatic Omicron breakthrough infection. They found that neutralizing antibodies targeting non-conserved RBD epitopes were specifically associated with protection from symptomatic infection, whereas cellular immunity was most effective when targeting conserved Spike epitopes. The authors concluded that achieving long-term protection against Omicron requires a balanced immune response to both conserved and non-conserved epitopes of the viral Spike protein. These findings have implications for future vaccine strategies and hold clinical significance. However, the manuscript requires significant improvements in structure, rationale, clarity, and presentation.

 Section 2.1. Study design

  • Line 78: Please provide reference for MOSAIC Study.
    • Study design need to be described in detail in the text, not in the figure legend.
    • Always define the acronym when first use, e.g., TP1 and TP2 in line 93.
    • Provide IRB as well as consent information.
    • Figure A1: Need annotations for icons in rows “asymptomatic N-seronegative” and “infected”.

Section 2.2. Blood collection and infection monitoring

  • Lines 107–108: Whole blood collected in lithium-heparin tubes was used for cell assays. What was the purpose of the blood collected in EDTA tubes?"
  • Lines 114 to 123: Please describe briefly what each test was for so that readers will not have to check by themselves. For example: Elecsys Anti-SARS-CoV-2 is an immunoassay intended for the qualitative detection of antibodies to SARS-CoV-2 in human serum and plasma. It uses a recombinant nucleocapsid (N) protein to detect antibodies against SARS-CoV-2. recomWell SARS-CoV-2 IgG and IgA are enzyme immunoassays used for the quantitative and qualitative detection of IgG and IgA antibodies, respectively, targeting the recombinantly produced SARS-CoV-2 N antigen.
  • Lines 117 to 118: If nasal and throat swabs were also used to detect viral infection, they should be included in the study design. Additionally, the title of section 2.2 should be changed to “Sample Collection” to reflect the broader scope of specimens collected.

Section 2.3. Antibody assays

  • Please describe briefly what the V-PLEX SARS-CoV-2, and V-PLEX SARS-CoV-2 (ACE2) assess for. For example:
    • V-PLEX SARS-CoV-2 Panel is a multiplex immunoassay designed to quantitatively measure IgG antibodies against multiple SARS-CoV-2 antigens (such as spike protein, RBD, and nucleocapsid) in human serum or plasma.
    • V-PLEX SARS-CoV-2 (ACE2) Panel assesses the inhibition of ACE2 binding to SARS-CoV-2 spike protein variants, serving as a surrogate neutralization assay to evaluate the functional activity of antibodies that block virus-receptor interaction.

Section 2.4. T cell assays

  • Lines 140 to 142: please clarify whether T cells or whole blood were stimulated with the peptides?
  • Briefly describe the T-Track SARS-CoV-2 Quant PCR, e.g., T-Track® SARS-CoV-2 Quant PCR is a quantitative PCR-based assay designed to assess the SARS-CoV-2-specific T cell response by measuring the expression of interferon-gamma (IFNG) and CXCL10 mRNA after in vitro stimulation of whole blood with SARS-CoV-2 antigens. It serves as an indirect measure of T cell activation and cellular immune response to the virus.

3. Results: The 1st paragraph should be in the methods.

3.1. Booster immunization does not result in sustained enhancement of Omicron RBD neutralization

  • Figure 1f: Make the title format consistent with the others by using a period ('.') instead of a comma (',').
  • Lines 232–235: The authors stated: “A decline in antibodies reacting with key variant epitopes involved in ACE2-binding, even if coupled with an increase of antibodies targeting conserved sites, may lead to a net loss of variant neutralizing capacity. This further suggests that a major fraction of the measured Omicron NAb inhibition at TP2 occurs at conserved sites.” Just wondering how the authors rule out a substantial decrease in Omicron targeting NAb production at TP2?
  • Additionally, the description of the RBD–ACE2 inhibition electrochemiluminescence immunoassay is unclear. It is not evident whether this assay measures inhibition, antibody concentration, or both. The authors should clarify this in the Methods section.

3.2. NAb levels are not generally suitable as correlates of protection

  • Overall, the only difference between the infected and uninfected groups was NAb inhibition at TP1 against the Delta variants. The authors should clarify which group showed higher NAb inhibition.
  • Please provide more rationale for gating on TP2 inhibition rather than TP1?
  • Are there differences in neutralizing antibody (NAb) inhibition at TP2 between infected and uninfected individuals who exhibit high NAb inhibition at TP2?

3.3. Loss of RBD-specific IgG antibodies after booster vaccination is associated with symptomatic infection

  • Figure 3 legend (Line 317): How were the Ab levels normalized against wild-type RBD?
  • The authors claim that 'loss of RBD-specific IgG antibodies after booster vaccination is associated with symptomatic infection.' It would be helpful to see the changes in Omicron-specific RBD and Spike IgG antibody levels from TP1 to TP2 in infected versus uninfected individuals.

 3.4. NAbs against non-conserved RBD epitopes are associated with protection from variant symptomatic infection

  • Not sure if wild-type - Omicron subvariants inhibition can represent inhibition against non-conserved RBD epitopes. Please explain more clearly.

3.5. Cellular immunity to conserved epitopes is associated with protection from symptomatic variant infection

  • Table A2: Spike B.1.617.2 mutation sites: 27 15mers Wuhan (wild type) p&e is in the table twice? Is that redundancy?
  • Lines 383-386: Cannot draw the conclusion from statistically insignificant data.

Overall:

  • The MNS requires restructuring. Some content currently in the Results section belongs in the Methods, while some results should be moved to the Discussion. Additionally, certain information in the figure legends should be included in the Methods.
  • Greater clarity and justification are needed for the selected methods and analytical strategies, such as the data gating approach.
  • The data presentation should be improved. Using correlation-style dot plots for multiple comparisons is confusing and should be reconsidered.
  • Since the manuscript proposes a potentially improved strategy for COVID-19 vaccines, the title should be revised to reflect this as a suggestion rather than a definitive claim.

Author Response

We thank you for taking the time to assess the manuscript and greatly appreciate your thorough and thoughtful comments and suggestions. The revision of the manuscript according to your suggestions has substantially improved its quality.

Section 2.1. Study design

  • Line 78: Please provide reference for MOSAIC Study.
    Author response: Results of the MOSAIC study have not yet been published. The current manuscript represents the first report from it. Further publications are planned to present additional findings. We have clarified this in Section 2.1 (page 3, line 87).
  • Study design need to be described in detail in the text, not in the figure legend.
    Author response: Details on the study design are now included in the text Section 2.1, (page 3, line 91-123). The figure legend has been shortened accordingly (Appendix A1, page20).
  • Always define the acronym when first use, e.g., TP1 and TP2 in line 93.
    Author response: TP1 and TP2 are now defined at first mention Section 2.1 (page 3, line 123 and line 106).
  • Provide IRB as well as consent information.
    Author response: In accordance with the journal’s format requirements, IRB and consent information are provided separately at the end of the main text (page 19, lines 596-598).
  • Figure A1: Need annotations for icons in rows “asymptomatic N-seronegative” and “infected”.
    Author response: Annotations are now included in Figure A1 (page 20).

Section 2.2. Blood collection and infection monitoring

  • Lines 107–108: Whole blood collected in lithium-heparin tubes was used for cell assays. What was the purpose of the blood collected in EDTA tubes?
    Author response: Thank you for pointing this out. During the revision, it was noted that EDTA plasma was erroneously mentioned in the manuscript instead of serum that had actually been used. Although EDTA tubes have indeed been collected during the MOSAIC study, results from EDTA tubes are not included in the manuscript. This has been corrected accordingly. Section 2.2 now includes the use of serum and heparin plasma (page 4, line 137-139).
  • Lines 114 to 123: Please describe briefly what each test was for so that readers will not have to check by themselves. For example: Elecsys Anti-SARS-CoV-2 is an immunoassay intended for the qualitative detection of antibodies to SARS-CoV-2 in human serum and plasma. It uses a recombinant nucleocapsid (N) protein to detect antibodies against SARS-CoV-2. recomWell SARS-CoV-2 IgG and IgA are enzyme immunoassays used for the quantitative and qualitative detection of IgG and IgA antibodies, respectively, targeting the recombinantly produced SARS-CoV-2 N antigen.
    Author response: A brief description of the assay principle is now provided in Section 2.2 (page 4, line 145-152).
  • Lines 117 to 118: If nasal and throat swabs were also used to detect viral infection, they should be included in the study design. Additionally, the title of section 2.2 should be changed to “Sample Collection” to reflect the broader scope of specimens collected.
    Author response: nasal/throat swabs are now mentioned in Section 2.1 (page 3, line 100-102). The title of Section 2.2 has been changed as recommended (page 4, line 136).

Section 2.3. Antibody assays

  • Please describe briefly what the V-PLEX SARS-CoV-2, and V-PLEX SARS-CoV-2 (ACE2) assess for. For example:
    • V-PLEX SARS-CoV-2 Panel is a multiplex immunoassay designed to quantitatively measure IgG antibodies against multiple SARS-CoV-2 antigens (such as spike protein, RBD, and nucleocapsid) in human serum or plasma.
    • V-PLEX SARS-CoV-2 (ACE2) Panel assesses the inhibition of ACE2 binding to SARS-CoV-2 spike protein variants, serving as a surrogate neutralization assay to evaluate the functional activity of antibodies that block virus-receptor interaction.
      Author response: A brief description of the assay principle is now provided in Section 2.3 (page 4, line 166-172).

Section 2.4. T cell assays

  • Lines 140 to 142: please clarify whether T cells or whole blood were stimulated with the peptides?
    Author response: The text in Section 2.4 is clearer now, specifying that stimulation was performed in whole blood (page 5, line 188-190).
  • Briefly describe the T-Track SARS-CoV-2 Quant PCR, e.g., T-Track® SARS-CoV-2 Quant PCR is a quantitative PCR-based assay designed to assess the SARS-CoV-2-specific T cell response by measuring the expression of interferon-gamma (IFNG) and CXCL10 mRNA after in vitro stimulation of whole blood with SARS-CoV-2 antigens. It serves as an indirect measure of T cell activation and cellular immune response to the virus.
    Author response: A brief description of the T-Track SARS-CoV-2 Quant PCR is now included in Section 2.4 (page 5, line 197-201).

3. Results: The 1st paragraph should be in the methods.
Author response: The first paragraph is now partially included in the methods section. The part with the epidemiological data has been moved to the introduction, as we consider it to be scientific background rather than material or methods (Section 1, page 2, line 66-75 and Section 2.1, page 3, line 87-123).

3.1. Booster immunization does not result in sustained enhancement of Omicron RBD neutralization

  • Figure 1f: Make the title format consistent with the others by using a period ('.') instead of a comma (',').
    Author response: Figure 1f has been corrected.
  • Lines 232–235: The authors stated: “A decline in antibodies reacting with key variant epitopes involved in ACE2-binding, even if coupled with an increase of antibodies targeting conserved sites, may lead to a net loss of variant neutralizing capacity. This further suggests that a major fraction of the measured Omicron NAb inhibition at TP2 occurs at conserved sites.” Just wondering how the authors rule out a substantial decrease in Omicron targeting NAb production at TP2?
    Author response: We recognize that the passage is a bit misleading. We just wanted to emphasize that, despite a marked difference between Wild-type and Omicron-NAb-inhibition, NAbs targeting conserved epitopes may still substantially contribute to the overall inhibition. As this topic is discussed in Section 4, we modified the passage (page 8, line 261-266).
  • Additionally, the description of the RBD–ACE2 inhibition electrochemiluminescence immunoassay is unclear. It is not evident whether this assay measures inhibition, antibody concentration, or both. The authors should clarify this in the Methods section.
    Author response: The intended use of the RBD–ACE2 inhibition electrochemiluminescence immunoassay is now described in Section 2.3 (page 4, line 169-172).

3.2. NAb levels are not generally suitable as correlates of protection

  • Overall, the only difference between the infected and uninfected groups was NAb inhibition at TP1 against the Delta variants. The authors should clarify which group showed higher NAb inhibition.
    Author response: Figure 2 now includes the median of infected/non-infected subjects at high or low wild-type TP2 Nab-inhibition (triangles) to visualize the differences between groups (including p-values as before).
  • Please provide more rationale for gating on TP2 inhibition rather than TP1?
    Author response: We improved the description of the rationale for gating (page 9, line 292-297).
  • Are there differences in neutralizing antibody (NAb) inhibition at TP2 between infected and uninfected individuals who exhibit high NAb inhibition at TP2?
    Author response: Figure 2 now includes the median of infected/non-infected subjects at high wild type TP2 Nab-inhibition (triangles). In addition, p-values for TP2-values are now included.

3.3. Loss of RBD-specific IgG antibodies after booster vaccination is associated with symptomatic infection

  • Figure 3 legend (Line 317): How were the Ab levels normalized against wild-type RBD?
    Author response: We recognize the misleading nature of this sentence. Ab levels have been min/max-normalized (as mentioned in the Methods section), they have not been normalized specifically against wild-type RBD. This is now clearly stated in the figure legend (page 11, line 351).
  • The authors claim that 'loss of RBD-specific IgG antibodies after booster vaccination is associated with symptomatic infection.' It would be helpful to see the changes in Omicron-specific RBD and Spike IgG antibody levels from TP1 to TP2 in infected versus uninfected individuals.
    Author response: We agree that changes in Omicron-specific RBD and Spike IgG antibody levels would be of interest. However, this would take us beyond the scope of this section. Our aim was to confirm the central role of RBD-specific antibodies in providing immune protection and that an immunologic shift from RBD- to Spike- (non-RBD) specific humoral immunity could therefore be associated with (Omicron and maybe other potential variant) infection. This immunologic shift is most clearly demonstrated by wild-type antigens, as they are identical to the immunogen. Using Omicron-specific antigens may inadequately reflect the true shift since RBD is a mutational hotspot, and altered binding to mutated sites may thus confound an apparent shift toward non-RBD Spike regions.

3.4. NAbs against non-conserved RBD epitopes are associated with protection from variant symptomatic infection

  • Not sure if wild-type - Omicron subvariants inhibition can represent inhibition against non-conserved RBD epitopes. Please explain more clearly.
    Author response: In section 3.4 we now present data as wild-type and mutant Nab-levels. For clarity, we moved the interpretation as conserved- and non-conserved-epitope reactivity to section 4. In section 4 we describe the rationale behind the concept of conserved- versus non-conserved epitope specificity (page 18, line 536-548).

3.5. Cellular immunity to conserved epitopes is associated with protection from symptomatic variant infection

  • Table A2: Spike B.1.617.2 mutation sites: 27 15mers Wuhan (wild type) p&e is in the table twice? Is that redundancy?
    Author response: The second peptide has been corrected from Wuhan (wild type) to B.1.617.2. They identify a pair of mutated site pools representing the mutated pool and the wild-type pool as reference (page 21).
  • Lines 383-386: Cannot draw the conclusion from statistically insignificant data.
    Author response: We believe this finding should be presented to ensure transparency of the overall data. We have moved the graphs to the Appendix (Figure A4, page 22) and revised the wording to emphasize the speculative nature of the conclusion (page 13, line 401-402).

Overall:

  • The MNS requires restructuring. Some content currently in the Results section belongs in the Methods, while some results should be moved to the Discussion. Additionally, certain information in the figure legends should be included in the Methods.
    Author response: Contents have been moved to appropriate sections of the manuscript as recommended.
  • Greater clarity and justification are needed for the selected methods and analytical strategies, such as the data gating approach.
    Author response: The description of the rationale for gating and methods have been improved as stated above.
  • The data presentation should be improved. Using correlation-style dot plots for multiple comparisons is confusing and should be reconsidered.
    Author response: We primarily used correlation dot plots to maximize data transparency. Although this approach increases visual complexity, we believe showing data distribution among study groups provides valuable information, particularly given small group sizes. To improve readability, we added median values and reduced the dot size in figure 2, allowing easy identification of central tendencies while preserving individual data points. Additionally, we moved figures 4i, 4j, 5a, and 5b to the appendix to distinguish key findings from supporting data.
  • Since the manuscript proposes a potentially improved strategy for COVID-19 vaccines, the title should be revised to reflect this as a suggestion rather than a definitive claim.
    Author response: The manuscript title has been revised as recommended to better reflect our key findings.

Reviewer 2 Report

Comments and Suggestions for Authors

In the present manuscript the authors report data suggesting that both conserved and non-conserved epitopes of the SARS-CoV-2 spike protein should be used in further vaccine development. The data obtained on 107 healthcare employees demonstrated the rearrangement of immune repertoire after vaccination and importance of a balanced immune response for long-term protection against variant infection.

Despite the fact that the development of anti-COVID vaccines is becoming less and less popular, the disease can still cause serious health issues, thus the results presented here are definitely important. 

However, prior to publication some minor corrections should be done 

  1. Please re-read the manuscript carefully and correct format where necessary (for example, lanes 88-89 should be re-formatted as “criteria: (1) no reported…; (2) negative…) 
  2. Lane 125 “The presence of anti- RBD and anti-Spike IgG antibodies..” Also, please check the overall correctness of this sentence. 
  3.  Please, in the supplement, specify the sequences (or at least AA positions) of the peptide used. Table A2 is almost useless.
  4. Lanes 175-192 should be removed from Results. Some of the information repeats information in materials and methods, while other is a kind of a background/introduction. 
  5. Same with lines 197-201; 373-382; 
  6. Figures 1 a-c - please enlarge legends (like in Fig. 2);

Overall, I suggest shortening the manuscript, reducing the amount of figures and making them more precise.

Author Response

We thank you for taking the time to assess the manuscript and greatly appreciate your thorough and thoughtful comments and suggestions. The revision of the manuscript according to your suggestions has improved its quality.

  1. Please re-read the manuscript carefully and correct format where necessary (for example, lanes 88-89 should be re-formatted as “criteria: (1) no reported…; (2) negative…)
    Author response: We have reformatted the manuscript as recommended (page 3, line 112-123).
  2. Lane 125 “The presence of anti- RBD and anti-Spike IgG antibodies..” Also, please check the overall correctness of this sentence.
    Author response: The sentence has been revised (page 4, line 162-166).
  3.  Please, in the supplement, specify the sequences (or at least AA positions) of the peptide used. Table A2 is almost useless.
    Author response: Detailed peptide sequences are not publicly available from all manufacturers. Order numbers are therefore now provided to allow researchers to obtain full product specifications directly from the supplier (table A2, page 21).
  4. Lanes 175-192 should be removed from Results. Some of the information repeats information in materials and methods, while other is a kind of a background/introduction. 
    Author response: The paragraph is now partially included in the methods section. The part with the epidemiological data has been moved to the introduction, as we consider it to be scientific background rather than material or methods (Section 1, page 2, line 66-75 and Section 2.1, page 3, line 87-123).
  5. Same with lines 197-201; 373-382;
    Author response: Sections have been integrated into section 2.3 (page 4, line 172-174) and 2.4 (page 5, line 185-208).
  6. Figures 1 a-c - please enlarge legends (like in Fig. 2);
    Author response: Figure legends are now enlarged.

Overall, I suggest shortening the manuscript, reducing the amount of figures and making them more precise.
Author response: Figures 4i, 4j, 5a, and 5b have been moved to the appendix to reduce the amount of figures in the manuscript. We improved the data presentation of Figure 2 by including median values and reduced the size of dots in Fig. 2, so that median values can easily be identified with data distribution being preserved. We believe that further substantial reduction of the manuscript length would be difficult to achieve without compromising essential data presentation.

Reviewer 3 Report

Comments and Suggestions for Authors

This is a very good example of careful research on SARS-CoV-2 vaccination, utilizing blood as the primary material and antibodies as a probe of neutralization. They also included a small set (11 patients) in cell response analysis, dealing with the response of vaccinated people against omicron variants. The main results concentrate on the analysis of antibodies in blood. /their results, despite extensive explanations, show the conclusion of the manuscript, which is an expected conclusion that "vaccination strategies that exclusively target either conserved or non-conserved epitopes may not be an effective means of establishing an adequate level of long-term protection against infection". These results are not surprising but deserve publication.

Author Response

Reviewer:

This is a very good example of careful research on SARS-CoV-2 vaccination, utilizing blood as the primary material and antibodies as a probe of neutralization. They also included a small set (11 patients) in cell response analysis, dealing with the response of vaccinated people against omicron variants. The main results concentrate on the analysis of antibodies in blood. /their results, despite extensive explanations, show the conclusion of the manuscript, which is an expected conclusion that "vaccination strategies that exclusively target either conserved or non-conserved epitopes may not be an effective means of establishing an adequate level of long-term protection against infection". These results are not surprising but deserve publication.

Author response:

We thank the reviewer for the time and effort invested in the review and would like to express our appreciation for the valuable feedback.

We have tightened some passages in the results section, removing unnecessarily extensive explanations to improve readability (section 3.1., page 6 line 240-241 and page 8, line 270-272; section 3.4. page 12 line 375; caption Figure 1 and 2).

Reviewer 4 Report

Comments and Suggestions for Authors

The authors found that vaccination strategies targeting only conserved or non-conserved epitopes are not an effective means of establishing a sufficient level of long-term protection against infection. In other words, they argue that balanced immunity against both conserved and non-conserved epitopes of the virus is necessary.

Furthermore, the authors argue that three things need to be met: (i) the second vaccination achieves sufficient antibody titers to trigger an efficient boost and repertoire rearrangement, (ii) the booster vaccination achieves sufficient antibody titers to prevent antibody fading, and (iii) immunity is balanced by providing sufficient antibody levels targeting a broad epitope repertoire including conserved and non-conserved epitopes.

As a reviewer, I think this is a very interesting and good study overall.

I would like to make only the following minor comments.

Figure 3 has a sufficient number, but Figure 5 has an overwhelmingly small number. I would like to hear a rebuttal that the data is still meaningful.

Author Response

Reviewer:

The authors found that vaccination strategies targeting only conserved or non-conserved epitopes are not an effective means of establishing a sufficient level of long-term protection against infection. In other words, they argue that balanced immunity against both conserved and non-conserved epitopes of the virus is necessary.

Furthermore, the authors argue that three things need to be met: (i) the second vaccination achieves sufficient antibody titers to trigger an efficient boost and repertoire rearrangement, (ii) the booster vaccination achieves sufficient antibody titers to prevent antibody fading, and (iii) immunity is balanced by providing sufficient antibody levels targeting a broad epitope repertoire including conserved and non-conserved epitopes.

As a reviewer, I think this is a very interesting and good study overall.

I would like to make only the following minor comments.

Figure 3 has a sufficient number, but Figure 5 has an overwhelmingly small number. I would like to hear a rebuttal that the data is still meaningful.

Author response:

Firstly, we would like to express our gratitude for your review of the manuscript and welcome your constructive feedback.

While we acknowledge the modest sample sizes (n=8 vs n=6), the detection of statistically significant differences despite these limitations indicates substantial effect sizes. Small samples inherently require large differences to achieve statistical significance, suggesting our observed effects are both statistically meaningful and likely clinically relevant. We believe that our results represent a meaningful addition to the evidence base that may inform larger scale investigation and guide future research directions in vaccine development. Although cellular data are not the primary focus of this study, they provide valuable insights suggesting that humoral and cellular immunity may exhibit differential dependence on specific epitopes for long-term immune protection within the same cohort.

Generally, standardized minimum sample size requirements have not been established for medical research studies. However, the value of significant findings from small-scale studies in advancing scientific knowledge is well established (see for example Billingham et al. Clin. Invest. (2012) 2(7), 655–657; Indrayan et al. J Postgrad Med 2021 Nov 26;67(4):219–223; Panos et al. Drug Des Devel Ther . 2023 Jul 3;17:1959–1961), particularly given resource constraints that characterized research conducted during the pandemic.